# Protocol for an observational cohort study investigating personalised medicine for intensification of treatment in people with type 2 diabetes mellitus: the PERMIT study

Patrick Bidulka [ID],[1] Stephen O'Neill,[2] Anirban Basu,[3] Samantha Wilkinson,[4] Richard J Silverwood [ID],[5] Paul Charlton,[6] Andrew Briggs,[2] Amanda I Adler,[7] Kamlesh Khunti,[8] Laurie A Tomlinson,[1] Liam Smeeth,[1] Ian J Douglas,[1] Richard Grieve[2]

For numbered affiliations see end of article.

**Correspondence to**
Mr Patrick Bidulka;
patrick.bidulka1@lshtm.ac.uk

## ABSTRACT

**Introduction** For people with type 2 diabetes mellitus (T2DM) who require an antidiabetic drug as an add-on to metformin, there is controversy about whether newer drug classes such as dipeptidyl peptidase-4 inhibitors (DPP4i) or sodium-glucose co-transporter-2 inhibitors (SGLT2i) reduce the risk of long-term complications compared with sulfonylureas (SU). There is widespread variation across National Health Service Clinical Commissioning Groups (CCGs) in drug choice for second-line treatment in part because National Institute for Health and Care Excellence guidelines do not specify a single preferred drug class, either overall or within specific patient subgroups. This study will evaluate the relative effectiveness of the three most common second-line treatments in the UK (SU, DPP4i and SGLT2i as add-ons to metformin) and help target treatments according to individual risk profiles.

**Methods and analysis** The study includes people with T2DM prescribed one of the second-line treatments-of-interest between 2014 and 2020 within the UK Clinical Practice Research Datalink linked with Hospital Episode Statistics and Office of National Statistics. We will use an instrumental variable (IV) method to estimate short-term and long-term relative effectiveness of second-line treatments according to individuals' risk profiles. This method minimises bias from unmeasured confounders by exploiting the natural variation in second-line prescribing across CCGs as an IV for the choice of prescribed treatment. The primary outcome to assess short-term effectiveness will be change in haemoglobin A1c (%) 12 months after treatment initiation. Outcome measures to assess longer-term effectiveness (maximum ~6 years) will include microvascular and macrovascular complications, all-cause mortality and hospital admissions during follow-up.

**Ethics and dissemination** This study was approved by the Independent Scientific Advisory Committee (20-064) and the London School of Hygiene & Tropical Medicine Research Ethics Committee (21395). Results, codelists and other analysis code will be made available to patients, clinicians, policy-makers and researchers.

## Strengths and limitations of this study

► This large representative study of UK clinical practice will describe variation in second-line antidiabetic treatment for people with type 2 diabetes mellitus.
► The instrumental variable (IV) design will minimise bias due to confounding by indication and provide person-level estimates of second-line treatment effectiveness.
► The IV design relies on assumptions which can only be partly tested using the data available.
► We will not consider less commonly used injectable second-line antidiabetic treatments in the UK, namely glucagon-like peptide-1 receptor agonists and insulin.

## INTRODUCTION

Around 3.5 million people in the UK have been diagnosed with type 2 diabetes mellitus (T2DM) accounting for ~10% of National Health Service (NHS) expenditure.[1] This proportion is predicted to rise to ~17% by 2035–2036.[2] T2DM is a progressive disease which requires careful management of blood glucose and diabetes-associated complications.[1] The National Institute for Health and Care Excellence (NICE) recommends metformin as the first-line antidiabetic treatment in people with T2DM.[1] In many cases, people with T2DM need further treatment in addition to metformin monotherapy to maintain sufficient glycaemic control.

NICE guidance recommends several drug classes as add-ons to metformin for first-stage intensification, hereafter referred to as second-line treatment. These include sulfonylureas (SU), pioglitazone, dipeptidyl peptidase-4 inhibitors (DPP4i) or sodium-glucose

co-transporter-2 inhibitors (SGLT2i).[1 3] NICE guidance recommends considering individual clinical circumstances when selecting T2DM drug treatment. For instance, SGLT2i are recommended as second-line treatment if the person is at high risk of hypoglycaemia or when SU are not tolerated or are contraindicated.[1 3] However, these guidelines do not specify a single preferred drug class, either overall or within specific groups sharing clinical characteristics.[1] Research using a representative sample of the UK primary care population up to 2017 showed that SU, DPP4i and SGLT2i, each in combination with metformin, are the most commonly prescribed second-line treatments.[4]

There is wide variation in the proportion of people prescribed these drugs in addition to metformin across NHS Clinical Commissioning Groups (CCGs), who commission local NHS services, suggesting clinician preference may influence treatment choice.[4] In particular, the variation in SGLT2i prescribing suggests that some clinicians may prescribe these drugs even for those patients who are not considered at high risk of hypoglycaemia and therefore eligible for SU.

Similar to NICE guidance, an international consensus statement published in 2018 did not specify a single preferred drug class for second-line treatment, but recommended that choice is 'personalised' to individual characteristics and risk profiles. This statement was updated in 2019 in light of new evidence supporting SGLT2i or glucagon-like peptide-1 receptor agonist (GLP1RA) use after metformin for those with atherosclerotic cardiovascular disease (CVD), or those at high CVD risk.[5] However, regulators, clinicians and patients remain uncertain about how best to tailor second-line antidiabetic treatment based on individual characteristics.

Meta-analyses have reported that compared with other antidiabetic treatments, second-generation or third-generation SU are not associated with higher risk of death or cardiovascular (CV) events.[6 7] A recent CV outcome trial reported that the safety profile of glimepiride (SU drug class) was similar to linagliptin (DPP4i drug class).[8] Several placebo-controlled trials reported that SGLT2i reduced major CV events in people with T2DM.[9–12] While head-to-head randomised controlled trials (RCTs) can provide unbiased estimates of relative effectiveness, the range and number of participants included in head-to-head RCTs of alternative second-line treatments are insufficient to provide reliable estimates of long-term effectiveness according to individual-level risk profiles.[13–15]

Observational studies comparing outcomes of alternative second-line drug regimens have reported that SU, DPP4i and SGLT2i combined with metformin are all associated with haemoglobin A1c (HbA1c) reductions compared with metformin alone[16 17]; however, some reported that SU are associated with higher risk of CV events.[7 18] These observational studies did not recognise that people who receive SU may have been a more severe case mix according to unmeasured prognostic variables (eg, frailty) meaning results are likely biased due to confounding by indication.[19]

## Aims and objectives

This study aims to investigate the relative effectiveness of SU, DPP4i or SGLT2i in combination with metformin as second-line antidiabetic drug treatments on key T2DM outcomes, and how treatment decisions should be tailored to an individual's risk factor profile to maximise clinical benefit. We will use advanced quantitative methods to minimise the impact of confounding by indication and allow for heterogeneity according to patient characteristics.

The study's objectives are to: (1) Describe baseline characteristics and treatment patterns overall, and by clinically important subgroups, for SU, DPP4i and SGLT2i in combination with metformin as second-line T2DM treatment; (2) Estimate the relative short-term (12 month) effectiveness of SU, DPP4i or SGLT2i combined with metformin on levels of HbA1c, overall and according to individual risk-factor profiles and (3) Estimate the long-term (maximum ~6 years) effectiveness of SU, DPP4i or SGLT2i combined with metformin on incident microvascular and macro-vascular complications, overall and according to individual risk-factor profiles.

## METHODS AND ANALYSIS

### Data resources

We will identify the study population using the UK Clinical Practice Research Datalink (CPRD),[20 21] a pseudonymised primary care database which includes detailed demographic/lifestyle data, clinical diagnoses and measurements, primary care prescriptions, referrals and laboratory test results for approximately 20% of the UK population. Both the CPRD Gold and Aurum datasets will be used to identify people eligible for inclusion, providing a representative population of people with T2DM eligible for the second-line treatments of interest.[20 21]

Linkage to Hospital Episode Statistics (HES) is available for approximately 70% of English practices and will be used to gather secondary care data for the study population. HES Admitted Patient Care data includes complete in-patient admissions data to all NHS hospitals in England.[22] These secondary care data include admission and discharge dates, diagnoses and other descriptive information (eg, ethnicity). Linkages will also be made to the Office of National Statistics to obtain mortality data and the Index of Multiple Deprivation (IMD) as a person-level proxy of socioeconomic status (SES).

### Sample selection/study population

The study population will include people registered with a CPRD-contributing practice, aged 18 years or older, diagnosed with T2DM who intensify antidiabetic treatment from metformin-monotherapy to a combination of metformin and SU, DPP4i or SGLT2i (second-line treatment) between 2014 and 2020 (figure 1).

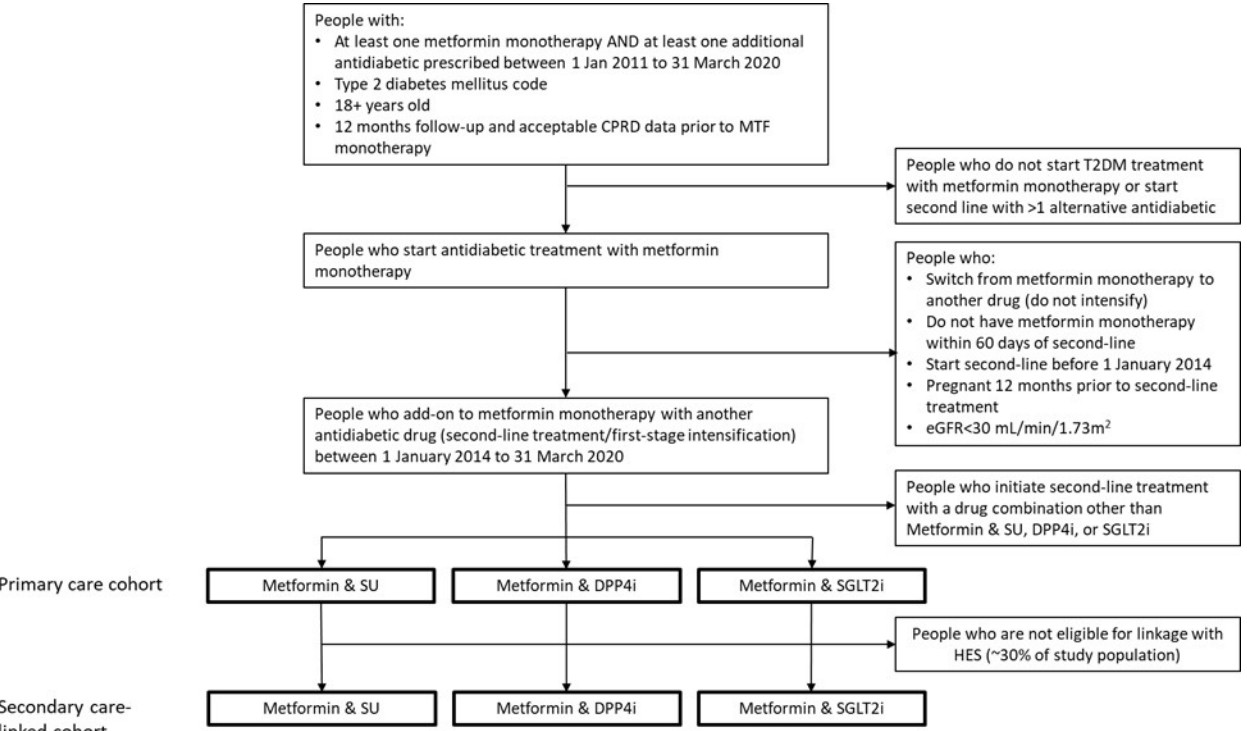

**Figure 1** Flow diagram illustrating the identification of the study cohort of people with type 2 diabetes mellitus (T2DM) who initiate second-line antidiabetic treatment with metformin and one of sulfonylurea (SU), dipeptidyl peptidase four inhibitor (DPP4i) or sodium-glucose cotransporter two inhibitors (SGLT2i). CPRD, Clinical Practice Research Datalink; eGFR, estimated glomerular filtration rate; HES, Hospital Episode Statistics; MTF; metformin.

We will identify people within CPRD with at least one prescription for metformin monotherapy and one other antidiabetic medication in primary care between 1 January 2011 and 31 March 2020, registered at a general practice (GP) contributing research-quality data at the prescription date, and registered with their GP for at least 1 year prior to the first metformin or other antidiabetic prescription, to ensure that we are studying new users. The study population will be limited to people with a T2DM primary care code, on or before the antidiabetic index date to exclude those prescribed antidiabetic medications for other indications (eg, polycystic ovarian syndrome or pre-diabetes). Using the individual's entire prescribing history with their registered GP, we will include only people who initiate antidiabetic treatment with metformin, and intensify metformin-monotherapy with a first-time prescription for one of the three second-line antidiabetic drug treatments of interest after 1 January 2014. We chose this date for the evaluation of these three treatments as prior to this only a small minority of people in the UK were prescribed SGLT2i.[4] People who intensify with two or more drug classes on the same date, who discontinue metformin monotherapy prior to a prescription for SU, DPP4i or SGLT2i, or who are prescribed a different drug class as second-line treatment (eg, thiazolidinedione (TZD), insulin, GLP1RA) will be excluded. In addition, we will exclude people with estimated glomerular filtration rate (eGFR) below $30\,mL/min/1.73m^2$ since SGLT2i are contraindicated for this group.[23] We will

also exclude women who were pregnant in the 12 months prior to second-line treatment initiation since prescribing guidelines recommend different treatments for pregnant and breastfeeding women.[24]

### Exposures

Exposure groups will include people prescribed SU, DPP4i or SGLT2i as an add-on to metformin. We will not consider other less commonly prescribed T2DM intensification treatments namely TZD, insulin and GLP1-RA since these treatments combined account for less than 10% of second-line therapy regimens in the UK.[4]

The first prescription date for second-line treatment will be considered baseline. To reduce misclassification of people who switch treatments rather than add-on to metformin, we require an additional prescription for metformin on the same date or within 60 days after the first prescription for the second-line drug prescription (SU, DPP4i or SGLT2i). This follows precedent research which used the same definition for second-line antidiabetic treatment in the same database.[4 25] The study will take an intention-to-treat approach, where each person will contribute to the original exposure group to which they were assigned, irrespective of which treatments they may be prescribed subsequently. People will remain exposed until the date the data are censored by death, the patient leaving the GP practice, the GP practice stops contributing to CPRD, or 31 July 2020. We will use the prescription duration recorded in primary care, plus a

60-day grace-period to account for stock-piling medicines, to mark the end of a prescription. Where these data are missing, we will impute the length of prescription with the mean duration of prescription at the practice level, plus the 60-day grace-period, to mark the end of a prescription. Data on subsequent anti-diabetic treatments (third line, fourth line, etc) will be described in those who discontinue second-line treatment.

The study requires information on each person's adherence to antidiabetic treatments. First, to provide a 'baseline' measure of adherence, we require a measure of each person's adherence to metformin monotherapy in the year prior to second-line treatment. We will then assess whether baseline adherence modifies the relative effectiveness of the alternative second-line treatments. Second, we will calculate treatment adherence during the follow-up to help interpret the estimates of the relative effectiveness of the alternative second-line treatments. For both measures of adherence, we will calculate defined daily dose (DDD) from the number of tablets and dosage instructions prescribed versus the duration of the period in question.

## Covariates

We will use primary care demographic data, diagnosis codes (Read or SNOMED for CPRD Gold and Aurum, respectively), and laboratory test results recorded prior to second-line treatment initiation, to define our main list of potential confounders. These include age, sex, IMD, time on first-line antidiabetic treatment (as a proxy for diabetes duration), GP size, relevant coprescriptions prescribed within 60 days of second-line treatment initiation (renin-angiotensin system inhibitors, statins), history of proteinuria and comorbidities at baseline (myocardial infarction (MI), unstable angina, stroke, ischaemic heart disease, hypoglycaemia, congestive heart failure (CHF), chronic kidney disease (CKD), end-stage renal disease (ESRD), cancer (any), advanced eye disease and lower extremity amputation). CKD status will be defined using serum creatinine test results to derive eGFR, using cutpoints defined by the Kidney Disease Improving Global Outcomes guidelines for CKD, but without requiring two measures 3 months apart.[26] We will also identify HbA1c, systolic blood pressure (SBP), diastolic blood pressure (DBP), eGFR, body weight and body mass index (BMI)[27] using values recorded in the 180 days period before the second-line antidiabetic treatment initiation date in the primary care record. Time between baseline clinical measures and second-line treatment initiation will also be included as covariates. We will follow previous observational research,[28] in undertaking secondary analyses that include additional potential confounders that are defined in primary care records, but for which we anticipate relatively high levels of missing data, namely: ethnicity, high-density lipoprotein (HDL), low-density lipoprotein (LDL), total cholesterol, triglycerides, smoking and alcohol status. In the HES-linked cohort, we will use International Classification of Diseases 10th Revision (ICD-10)

diagnosis codes recorded as part of previous hospitalisations in conjunction with primary care data to define comorbidities. We will also use ethnicity recorded in HES for people whose ethnicity is missing within the primary care data. Codelists for all covariates defined in primary and secondary care will be published alongside study results.

## Outcomes

The primary outcome for objective 2 (short-term relative effectiveness) will be absolute change in HbA1c% at 12 months follow-up. This change in HbA1c% will be quantified by contrasting follow-up versus baseline laboratory test data recorded in CPRD for each exposure. Secondary outcomes that will also be reported at 12 months after baseline include HDL, LDL, total cholesterol, triglycerides, SBP, DBP, eGFR, body weight and BMI. In defining the 12-month follow-up measurement, the available measure that is closest in time to the 12 months from baseline will be used, recognising that within the pilot data, a median interval in HbA1c measurement of 5 months was observed. Patients without the relevant measurement between 9 and 15 months will be designated as having 'missing 12-month data' (see missing data section). We will also report change in HbA1c at 6–18, 24–30 and 36 months follow-up, again using the closest HbA1c measure in the 3 months before and after the follow-up time point of interest.

Outcomes for long-term relative effectiveness (objective 3) will include macrovascular and microvascular conditions such as CV outcomes (MI, CHF, unstable angina, stroke), renal outcomes (nephropathy, ESRD, 40% decline in eGFR from baseline[29]) and lower limb amputation. Additional outcomes will include hypoglycaemia, time-to-cessation of second-line treatment or treatment switching, adherence calculated according to DDD, all-cause mortality and number of hospital admissions (any reason).

The assessment of long-term outcomes will use the maximum available follow-up. The investigation of microvascular and macrovascular complications including any hospitalisations will require HES data linked to CPRD, and so the patients in the CPRD cohort who cannot be linked to HES (an expected 30%–40%) will be excluded from this aspect of the evaluation.[20 21] Hospital admissions, including microvascular and macrovascular complications, will be identified in HES using ICD-10 diagnosis codes. Clinical diagnoses in primary care coded using Read (CPRD Gold) or SNOMED (CPRD Aurum) codes will be used in addition to secondary care data to identify outcome events. eGFR will be calculated using serum creatinine recorded in primary care as an input in the CKD-EPI formula.[30] We will define nephropathy as new-onset albuminuria or eGFR $<60 \,mL/min/1.73m^2$ in people with eGFR $\geq 60 \,mL/min/1.73m^2$ and no raised albumin to creatinine ratio within 2 years of second-line treatment initiation. A 40% decline in eGFR will be defined as an eGFR measure $\leq 40\%$ of baseline eGFR.

ESRD will be identified by primary care coding for ESRD and/or renal replacement therapy (RRT) by the GP.

## Analytical approach
### Objective 1: Describing UK treatment patterns for second-line T2DM treatment, and summarising the results of relevant published RCTs to contextualise the study findings
We will describe trends in prescribing for T2DM second-line treatment for the duration of the study period across the UK and between CCGs. This analysis will update previous research which described the same second-line treatment use in the UK from 2000 to 2017, and will employ similar methods.[4] These descriptive statistics will inform the assessment of the validity of the assumptions that underlie the overall study design. Baseline characteristics listed in the covariates section will also be described for this cohort, overall and stratified by exposure group. We will also conduct a literature review to summarise published RCTs which describe the relative effectiveness of alternative second-line antidiabetic treatments of interest to this study. This will help contextualise the results of this observational study (cf. objectives 2 and 3). We will consider reasons for any possible differences between this observational study compared with published RCTs, including residual confounding and differences in the study populations.

### Objectives 2 and 3: Instrumental variable (IV) design to estimate relative treatment effectiveness overall and by subgroup
Studies which apply traditional risk adjustment approaches with little information on case severity may provide biased estimates of treatment effectiveness. We will therefore use an IV design[31 32] to estimate treatment effectiveness in the presence of residual confounding. The IV for second-line drug treatment in this study will be each CCG's prescribing history, recognising that the choice of second-line treatment may involve the hospital diabetologist, the GP, other healthcare professionals, and the individual. We will define 'CCG prescribing history' as the proportion of people prescribed each second-line treatment in the CCG for the last complete calendar year prior to the treatment intensification currently under consideration. This IV encourages receipt of the

treatment but does not have a direct effect on outcomes except through the treatment prescribed (figure 2). Using CCG prescribing history as the IV follows pharma-coepidemiological research[32] that uses provider preference as an instrument for treatment prescribed.

In our pilot CPRD data,[4] the proportions of people prescribed each second-line treatment regimen varied widely. For example, in 2014 the ranges across CCGs were 5%–100% (SU), 0%–90% (DPP4i) and 0%–35% (SGLT2i).[4] These proportions have changed over time but similar people received different second-line treatment regimens simply according to CCG prescribing preference or time period.[4]

This study's design will exploit this wide variation in the choice of second-line treatment. We will use this IV to estimate the relative effectiveness of alternative second-line treatments while minimising bias from unobserved confounding. We will use a 'local IV estimator'[33] to allow for heterogeneity according to unobserved characteristics (eg, lifestyle choices) as well as observed characteristics (eg, baseline HbA1c) when reporting the relative effectiveness of the alternative second-line treatments according to individual risk factor profiles.

### IV assumptions
The validity of our IV design relies on three key assumptions: the IV must (1) strongly predict the treatment prescribed; (2) be independent of baseline unmeasured covariates; and (3) only affect the outcome through the treatment prescribed.[31] The IV design will lead to bias if the prescribing history of the CCG has a direct effect on the outcome. We carefully assessed whether the CCG's prescribing history met the criteria for an IV. Our pilot data showed it was strongly associated with the second-line treatment regimen prescribed (assumption 1). We also found that prescribing history balanced the observed covariates (assumption 2, figure 3). We are unable to assess empirically whether clinicians' prescribing history is independent of unmeasured confounders; however, it is likely that participants will attend their local GP without considering their prescribing history, and unlikely that the CCGs prescribing history would have a direct effect on outcomes (assumption 3). For example, it is unlikely that simply because a CCG shows a preference for prescribing SU the participants' outcomes would be better (or worse) regardless of the treatment actually prescribed. We will reassess each assumption using the full study dataset and undertake sensitivity analyses to test these assumptions.

### Power considerations
Power calculations were conducted prior to accessing study data. Clinically meaningful between-treatment difference in HbA1c from baseline is considered to be 0.3 percentage points (eg, from 8.0% to 7.7%) by the European Medicines Agency[34] and 0.5 percentage points by NICE.[35] We based our power calculations on these numbers and assuming an SD of 2.4.[36] We follow methodological recommendations for power calculations with

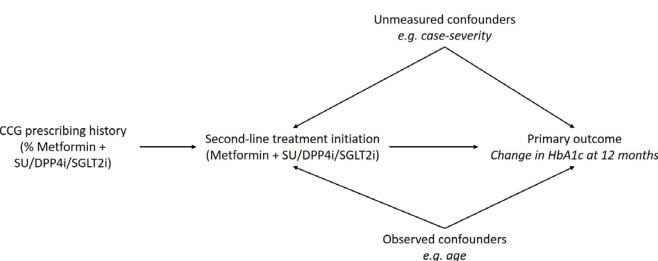

**Figure 2** Instrumental variable design to be applied in this study comparing three options for second-line antidiabetic treatment. CCG, Clinical Commissioning Group; DPP4i, dipeptidyl peptidase-4 inhibitors; HbA1c, haemoglobin A1c; SGLT2i, sodium-glucose cotransporter-2 inhibitor; SU, sulfonylureas.

These demonstrate that clinician choice of different first-stage intensification therapies show remarkably little relationship with clinical risk factors, strongly suggesting the IV approach will produce balanced and valid comparison groups.

(A) Preference for DPP4i prescription (prescriptions of DPP4i excluding current participant / total prescriptions) by year and CCG.

(B) Preference for SGLT2i prescription (prescriptions of SGLT2i excluding current participant / total prescriptions) by year and CCG.

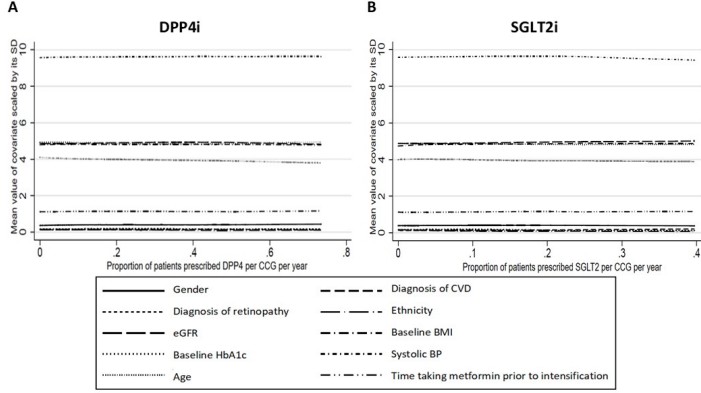

**Figure 3** Covariate balance across levels of CCG prescribing history (this study's IV) (2014–2017). BMI, body mass index; BP, blood pressure; CCG, Clinical Commissioning Group; CVD, cardiovascular disease; DPP4i, dipeptidyl peptidase-4 inhibitors; eGFR, estimated glomerular filtration rate; HbA1c, haemoglobin A1c; IV, instrumental variable; SGLT2i, sodium-glucose cotransporter-2 inhibitor; SU, sulfonylureas.

IV designs and consider that the proportion of people who actually receive the treatment predicted by the IV is 80%, but also consider scenarios where the IV is weaker (70% compliance) and stronger (90% compliance).[31] We require 80% power at the 5% (two-sided) level of statistical significance, with a Bonferroni correction to account for multiple comparisons to get a familywise error rate of 5%. Table 1 shows the requisite sample sizes of the two treatment groups projected to have the fewest participants (SU and SGLT2i). The study will include approximately 25 700 participants (SU=6000, DPP4i=13 000, SGLT2i=6700) based on an initial feasibility count, which will be more than sufficient for detecting whether clinically significant differences in the primary endpoint are statistically significant.

## Planned analyses

We will examine the relevant trends in prescribing between 2014 and 2020 by CCG, and by year, and summarise baseline covariates using data collected prior to the index date for second-line treatment.

We will provide personalised estimates of treatment effectiveness using the local IV (LIV) approach[33 37] to predict the counterfactual outcomes that each person would experience if they were prescribed each second-line treatment. We will use probit regression models[38] to estimate the propensity to receive each treatment according to observed characteristics, and CCG preference for each second-line regimen (the IV). We will estimate the relationship of each outcome with observed characteristics, and the propensity for each second-line treatment using generalised linear models (GLMs)[39] for continuous and count outcomes. For time-to-event outcomes (eg, time to second-line treatment cessation, time to each microvascular or macrovascular complication), we will recognise that the period of observation may differ across individuals due to censoring. We will describe each endpoint by plotting Kaplan-Meier curves, and estimate each treatment effect using discrete-time hazard models.[40] SEs will be calculated with non-parametric bootstrapping, and will account for clustering of individuals within practices.

These models will be used to estimate the relative effect of prescription of SGLT2i vs SU, DPP4i versus SU and SGLT2i vs DPP4i for the primary and secondary outcomes. Person-level treatment effects will be calculated as the difference

**Table 1** Required sample size (N) for the IV design according to instrument strength (level of compliance) and magnitude of effect size at 80% power and 5% (two-sided) level of statistical significance

| | Level of compliance (IV strength) | | | | | |
| --- | --- | --- | --- | --- | --- | --- |
| | 70% | | 80% | | 90% | |
| Effect size: between-treatment difference in mean HbA1c reduction baseline to 12 months | SU | SGLT2i | SU | SGLT2i | SU | SGLT2i |
| 0.3 | 4556 | 1952 | 3488 | 1495 | 2756 | 1181 |
| 0.4 | 2563 | 1098 | 1962 | 841 | 1550 | 664 |
| 0.5 | 1640 | 703 | 1256 | 538 | 992 | 425 |

DPP4i, dipeptidyl peptidase-4 inhibitors; IV, instrumental variable; SGLT2i, sodium-glucose cotransporter-2 inhibitor; SU, sulfonylureas.

in predicted outcomes following prescription of the alternative drugs. These person-level treatment effects will be aggregated to report the relative effectiveness of the treatments prescribed overall, and by prespecified subgroups. These prespecified subgroups will include: people with and without CV comorbidities overall and by subtype of CVD, people with baseline eGFR ≥60 mL/min/1.73 m[2] vs baseline eGFR <60 mL/min/1.73 m[2], age groups, sex, ethnicity, BMI (based on WHO categorical definition[41]), adherence to metformin and baseline HbA1c levels. We will consider finer eGFR subgroupings, age categories and HbA1c levels based on descriptive statistics (objective 1) prior to any relative effectiveness analyses (objectives 2–3). Any additional subgroups will be informed by descriptive statistics of each covariate, and the advice of a panel of healthcare professionals, building on those identified in a literature review. The clinical panellists will include diabetologists, GPs and practice nurses involved in care for people with T2DM.

## Missing data

In our primary analysis, we will use a complete-case approach based on the main potential confounders listed in the covariates section. We will conduct secondary analyses using complete cases for the full list of potential confounders, including those expected to have a high proportion of missing data (see covariates section), which we do not expect to be missing at random. Because we cannot assume covariate measurements are missing at random and the IV model is computationally intensive, we will not use multiple imputation.

We will adopt two main approaches based on the type of missingness for outcome data: (1) linear interpolation using values recorded during follow-up, and (2) inverse probability weighting (IPW) to those people lost to follow-up with no subsequent outcome measure. We will use linear interpolation for values that are intermittently missing during follow-up, for example, if HBA1c at 12 months is required, but the available measures are at 3-month and 17-month follow-up, which fall outside the requisite time window (9–15 months). This method was used in precedent diabetes research with observational data.[42] For those settings, were the patient is lost to follow-up and there is therefore no subsequent HbA1c measure available, we will use IPW, reweighting the data for those with available observations to represent the group lost to follow-up, assuming therefore that the HbA1c data are missing at random.[42 43]

## Sensitivity analyses

We will conduct sensitivity analyses falling under three broad categories: (1) Modified study population inclusion and exclusion criteria to evaluate the validity of our IV assumptions in subgroups where there is arguably less equipoise in the choice of second-line treatment[44]; (2) Comparing the larger primary care unlinked cohort with the primary care population linked to secondary care data (HES) and (3) Evaluating the robustness of our statistical methods.

Under the first category of sensitivity analyses, we will exclude people with contraindications for SU (eg, liver disease) who are prescribed SGLT2i, as this prescribing may not be due to CCG preference. We will also expand the eGFR exclusion to all those with eGFR <60 mL/min/1.73 m[2] (vs eGFR <30 mL/min/1.73 m[2] in the main analysis). In addition, we will include people who are censored or die during the first 60 days after a prescription for SU, DPP4i or SGLT2i without a prescription for metformin in the same time period to consider the impact of potentially misclassifying these people as switching from metformin monotherapy instead of adding on to metformin monotherapy. Under the second category of sensitivity analyses, we will repeat the analyses for objectives 1 and 2 limited to the HES-linked subpopulation from CPRD who are eligible for the long-term outcomes (objective 3). Third, we will assess the robustness of findings to alternative statistical models for the LIV approach, outcome regressions and alternative approaches to handling missing data.[45]

## Patient and public involvement

Two PP representatives were consulted when designing this study prior to obtaining funding. One has close family experience of type 1 diabetes as a carer and the other was recently diagnosed with T2DM. Both PP representatives have discussed the study with local patients and obtained future workshop interest. Our PP representatives reinforced that the study design, outcomes and interpretation should recognise the importance of personalising treatment choice according to the individual's experience, and according to their age, weight, ethnicity and more general lifestyle choices. The PP representatives have supported plans for two study workshops that will inform the translation of results to patients and the public. The PP representatives have emphasised the importance of developing accessible preworkshop information to help participants prepare. The PP representatives will help inform the way the study presents and communicates results so they are accessible to patients and the general public.

## Strength and limitations

This study will exploit the natural variation in prescribing patterns for second-line antidiabetic treatment across CCGs within similar groups of people by using an IV study design. This design minimises potential biases resulting from unmeasured confounders, a major limitation in observational research. The large and representative sample from UK primary care will improve the generalisability of this study's results and allow for stratification on prespecified baseline risk factors, helping patients and their providers choose treatments based on personal risk profiles to maximise clinical benefit. While the IV relies on three major assumptions which may limit the validity of our estimates, we will evaluate the strength of our assumptions in sensitivity analyses.

A potential limitation is that the required natural variation in prescribing may not exist for those people who are prescribed SGLT2i as second-line treatment because they do not tolerate or have a contraindication for SU, as per current[6]

NICE guidelines.[3] We will investigate this potential source of bias by undertaking a sensitivity analysis excluding those in the SGLT2i exposure group with contraindications for SU. In addition, our study may be susceptible to non-differential outcome misclassification, as we are unable to link our data to additional audit datasets with more detailed outcome information (eg, laboratory tests) such as the Myocardial Ischaemia National Audit Project (MINAP).[46] However, a previous study shows that the majority of MINAP acute MI events in the general England and Wales populations are also recorded in CPRD and HES.[47]

## Future work

The results of this study will be used in future research which aims to predict long term outcomes and associated costs to the NHS beyond this study's maximum follow-up. To do this, we will adapt a diabetes microsimulation model developed using observational data from the United States Veterans' Affairs database[48] to the UK setting. We will use a personalised approach to second-line treatment by using the estimates of relative effectiveness within the subgroup analyses in this study. We will publish the analysis plan for this work separately.

## Ethics and dissemination

### Ethics

This study will be based in part on data from the CPRD obtained under licence from the UK Medicines and Healthcare products Regulatory Agency. The data are provided by patients and collected by the NHS as part of their care and support. The interpretation and conclusions contained in this protocol are those of the authors alone. The study was approved by the Independent Scientific Advisory Committee (approval number 20-064) and the London School of Hygiene & Tropical Medicine Research Ethics Committee (reference number 21395). GPs have opted-in to contributing data to the CPRD, while individuals registered at these GPs may opt-out. Individual-level consent was not necessary since these data are deidentified.

### Dissemination/outputs

This study's outputs will be designed in collaboration with our expert advisory panel and PP representatives to help ensure this study can inform future clinical guidelines and care for people with T2DM. Results will be published open-access in peer-reviewed journals and presented at scientific conferences. Additional emphasis will be placed on the implementation of advanced quantitative methods in this study, which will provide general guidance for future studies on how the overall approach of combining these methods with routinely available electronic health data can provide insights to inform person-level care. We will provide recommendations via the Academic Health Sciences Networks to commissioners and T2DM care providers on how to target second-line antidiabetic treatment to individuals and patient groups.

Data visualisations of key results and lay summaries will also be published on this website as a resource to be shared with key stakeholders and as accessible information for the general public.

**Author affiliations**
[1]Department of Non-Communicable Disease Epidemiology, London School of Hygiene and Tropical Medicine, London, UK
[2]Department of Health Services Research and Policy, London School of Hygiene and Tropical Medicine, London, UK
[3]The Comparative Health Outcomes, Policy & Economics (CHOICE) Institute, University of Washington School of Pharmacy, Seattle, Washington, USA
[4]Personalized Healthcare Data Science, Roche Products Limited, Welwyn Garden City, UK
[5]Centre for Longitudinal Studies, University College London, London, UK
[6]Patient Research Champion Team, National Institute for Health Research, Twickenham, UK
[7]Diabetes Trials Unit, The Oxford Centre for Diabetes, Endocrinology and Metabolism, University of Oxford, Oxford, UK
[8]Diabetes Research Centre, University of Leicester, Leicester, UK

**Acknowledgements** KK is supported by the National Institute for Health Research (NIHR) Applied Research Collaboration East Midlands (ARC EM) and the NIHR Leicester Biomedical Research Centre (BRC).

**Contributors** PB and RG drafted the manuscript. SO'N, AB, SW, RJS, PC, AB, AIA, KK, LAT, LS and IJD assisted in the revision of the draft paper. Each of the authors has approved the final version of the manuscript for submission.

**Funding** This work was supported by National Institutes of Health Research (NIHR) grant number NIHR128490.

**Disclaimer** The views expressed are those of the author(s) and not necessarily those of the NIHR, NHS or the Department of Health and Social Care.

**Competing interests** PB, SO'N, AB, RJS, PC, LAT and LS have nothing to declare. SW is employed by Roche and holds stock in Roche. AB is an economic advisor on DiRECT trial with ongoing responsibility for economic analysis during long-term follow-up phase, and has also acted as consultant to GlaxoSmithKline, Merck, Novo Nordisk and Boehringer Ingelheim in relation to their diabetes products. AIA receives salary from the National Institute for Health Research (NIHR) via the University of Oxford and Addenbrooke's Hospital, and also chairs an NICE technology appraisal committee, is a member of Diabetes UK, and manages people whose salaries are partially funded by completed trials involving sitagliptin and exenatide and an ongoing trial of empagliflozin. KK has acted as a consultant, speaker or received grants for investigator-initiated studies for Novartis, Novo Nordisk, Sanofi-Aventis, Lilly and Merck Sharp & Dohme, Boehringer Ingelheim, Bayer, Berlin-Chemie AG/Menarini Group, Janssen, and Napp. IJD holds an unrestricted research grant from GSK and holds shares in GSK. RG sits on the NIHR commissioning committee.

**Patient and public involvement** Patients and/or the public were involved in the design, or conduct, or reporting, or dissemination plans of this research. Refer to the Methods section for further details.

**Patient consent for publication** Not applicable.

**Provenance and peer review** Not commissioned; externally peer reviewed.

**ORCID iDs**
Patrick Bidulka http://orcid.org/0000-0001-7644-2030
Richard J Silverwood http://orcid.org/0000-0002-2744-1194

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
