## [Reviewer comments · BMJ Open]

ARTICLE DETAILS

TITLE (PROVISIONAL)	A protocol for an observational cohort study investigating PERSONALISED Medicine for Intensification of Treatment in people with type 2 diabetes mellitus: the PERMIT study
AUTHORS	Bidulka, Patrick; O'Neill, Stephen; Basu, Anirban; Wilkinson, Samantha; Silverwood, Richard; Charlton, Paul; Briggs, Andrew; Adler, Amanda; Khunti, Kamlesh; Tomlinson, Laurie; Smeeth, Liam; Douglas, Ian; Grieve, Richard

VERSION 1 – REVIEW

REVIEWER	Bertoluci, Marcello C. Universidade Federal do Rio Grande do Sul, Endocrinology
REVIEW RETURNED	06-Feb-2021

GENERAL COMMENTS	The present submission is a protocol for developing a large population observational study (cohort) using screening data from the UK Clinical Practice Research Datalink (CPRD) from 2014 to 2020. The main objective is to compare the effectiveness of using a second glucose-lowering agent on top of metformin for adults with T2DM. The study has 3 objectives: 1) To describe baseline characteristics patterns of treatment by subgroups, for sulphonylureas (SU), DPP4i, and SGLT2i in combination with metformin as second line T2DM treatment; 2) Estimate the relative 12 month effectiveness of SU, DPP4i, or SGLT2i, plus metformin on levels of HbA1c, and searching for associations with individual risk-factor profiles; and 3) Estimate the long-term effectiveness of SU, DPP4i, or SGLT2i combined with metformin on incident micro- and macro-vascular complications. - The idea of the study is valid and sound because the propose to study the variables that may influence prescription of these 3 drugs besides metformin in UK. However I have some observations and questions: Methods: - Exposure The exposure groups will include people prescribed SU, DPP4i, or SGLT2i as an add-on to metformin. Why not to include people with GLP1-RA? And insulin? - Covariates: How economic status will impact in effectiveness? How will low-income people's prescription be addressed, considering higher costs for ISGLT2 and DPP4I? Will it be a bias? Outcomes: - HbA1c varies a lot during 12 months. How many measurements will be considered per patient. Will you determine it prospectively? - Cardiovascular outcomes need to be better defined. How an acute myocardial infarction or stroke will be registered? Will image
--

	methods be used? It is not clear if you will search for it prospectively or look into medical registry.  - Statistical methods should be better detailed for the baseline clinical and lab characteristics. - The authors will use instrumental variable design method, a sophisticated method to account for unexpected behavior between variables, and to estimate relative treatment effectiveness. - Sensitivity analysis are well described. - Missing data are well addressed. - A concern is how to deal with patients who may change from one group to another (eg when a certain medication is momentarily out in the market or may not available and the physician just change. - A major concern is that T2DM patients generally receive SU when in a more advanced T2DM stage with longer duration, when they become more insulinopenic. With a longer duration. In these patients, however, we also expect a higher incidence of cardiovascular disease. By this way, observational studies may face a reverse causality effect, falsely increasing cardiovascular risk for SU. How to deal with this?
--	--

REVIEWER	Golubnitschaja, Olga Friedrich-Wilhels-University of Bonn
REVIEW RETURNED	10-May-2021

GENERAL COMMENTS	The paper is of great clinical interest presenting valuable data. However, the below proposed revisions may significantly increase visibility and overall quality of this important publication.  1. Keywords should be extended presenting items highly relevant for the study which would attract more attention of multi-professional groups. Following items might be helpful: Patient stratification, disease modelling, targeted treatment prediction. 2. Statements in the "Introduction" should be updated keeping in mind a broad spectrum of T2DM related secondary pathologies and corresponding economic burden to healthcare. In particular, the 3rd sentence should be completed such as (just a proposal) "...careful management of blood glucose and cascading pathologies". Corresponding references should be provided - see some suggestions presented under the below point 4. 3. Legends should be essentially extended to detail the clinical relevance of the presented data. 4. References should be essentially updated: in the current version of the manuscript only 2 references originated from the year 2020 and no one from the year 2021. Below proposed items and corresponding references might be useful to fulfil the task by completing "Introduction", "Discussion", "Future work" as well as "Abstract":  A. Individualised Prediction and Targeted Prevention in diabetes care:  - Nomogram prediction for the 3-year risk of type 2 diabetes in healthy mainland China residents. 2019. doi: 10.1007/s13167-019-00181-2. - Suboptimal health status as an independent risk factor for type 2 diabetes mellitus in a community-based cohort: the China suboptimal health cohort study. 2019. doi: 10.1007/s13167-019-0159-9.
--

	B. Tools for companion diagnostics, disease prediction and prognosis (Big data, multi-omics, AI, Machine learning, Biobanks):  - Cell-free nucleic acid patterns in disease prediction and monitoring-hype or hope? 2020. doi: 10.1007/s13167-020-00226-x. - Biobanks in the era of big data: objectives, challenges, perspectives, and innovations for predictive, preventive, and personalised medicine. 2020. doi: 10.1007/s13167-020-00213-2 - Artificial intelligence supported patient self-care in chronic heart failure: a paradigm shift from reactive to predictive, preventive and personalised care. 2019. doi: 10.1007/s13167-019-00188-9. - Evaluation of machine learning methodology for the prediction of healthcare resource utilization and healthcare costs in patients with critical limb ischemia-is preventive and personalized approach on the horizon? 2020. doi: 10.1007/s13167-019-00196-9. 5. Keeping in mind the above listed points, "Abstract " should be thoroughly elaborated presenting  A. DM relevant UK / global statistics which would clearly support the study objectives, design and conclusions B. concluding statements in the context of predictive and personalised medicine to improve DM related primary, secondary and tertiary care C. outlook (predictive models, AI, cost-efficacy, improved life quality of the patients).
--	---

REVIEWER	Samocha-Bonet, Dorit Garvan Institute of Medical Research, Diabetes Division
REVIEW RETURNED	25-May-2021

GENERAL COMMENTS	This study aims to evaluate the relative effectiveness of the three most common second-line medications added on to the first-line metformin in the UK. Currently SU, DPP4i and SGLT2i are added based on the physician discretion/preference in individuals with type 2 diabetes who do not reach glycaemic target on metformin alone and this study aims to improve the tailoring of the added medication to the participant's risk profile using the Instrumental Variable method. The study aims are to:  1. Describe the current treatment patterns overall, and by clinically important subgroups in the UK 2. Estimate the relative short-term (12 months) effectiveness of the add on medications on HbA1c (primary endpoint) 3. Estimate the long-term (maximum 6 years) effectiveness on micro- and macro- vascular complications, overall and according to individual risk-factor profiles. Overall, I feel that I did not fully understand how will this study improve the treatment of individuals with type 2 diabetes. Do the authors suggest that the IV method will be superior to the current method (physician preference)? Apologies if I got it all wrong, but as someone who reads the literature and researches this area themselves, it probably means that further details and clarifications should be added. Methods
--

	1. Please clarify how the clinically important subgroups will be determined. Are they predefined based on the baseline data? 2. The choice to include individuals as young as 18 year old, would limit the possibility to assess the micro- and macro-vascular benefits of the treatment. Please explain why you did not pick an older cut-off. 3. The inclusion of individuals treated with metformin from 2011 to 2020 means that some will have longstanding diabetes and some will be recently diagnosed. Is this a limitation? 4. Regarding the sample size/power calculation: a. Why does Table 1 not include the sample size for the DPP4i? b. Please explain why the sample size (27,000) is several folds higher than the minimum required (for 80% power and effect size of 0.4% reduction, Table 1). Is this to account for expected loss to follow up?
--	--

REVIEWER	Nilsson, Peter Lund University, Clinical Sciences
REVIEW RETURNED	25-May-2021

GENERAL COMMENTS	This is description of the design of an observational study comparing the prescription patterns of second-line drugs as add-on to metformin in patients with type 2 diabetes in UK primary health care. Even if the design is purely observational, new statistical methods (instrumental variable = local recommendations by authorities) are introduced to minimize the risk of confounding by indication. Three classes of second-line anti-diabetes drugs will be analysed for prescription patterns during 2014-2020 in different regions of the UK (SU, DPP-4 inhibitors; SGLT-2 inhibitors), but not GLP-1 receptor agonists/analogues because much less prescription in clinical practice during early years. I think that the aim and methods of this observational study are mostly satisfactory, but a first focus on control of HbA1c over a 12 months period is not enough. What matters is the potential to prevent hard clinical endpoints (CVD and renal) as well as mortality patterns and hospitalisations (available in HES for 70% of all hospitalisations) during a 6 year period, as HbA1c is more linked to microvascular endpoints than to macrovascular endpoints. It is not clear whether prevention of nephropathy is included as an endpoint or not, just "microvascular endpoints" are mentioned. The authors state that they will consider ESRD, and 40% decline in eGFR from baseline. However, nephropathy is not explicitly mentioned. A number of covariates are listed, but not including BMI. There might exist a bias in prescription patterns according to BMI of the patient. It is well-known that SU will increase hyperinsulinaemia as often found in obese patients with underlying insulin resistance. Why is not BMI considered? Only as a secondary outcome at 12 months is BMI considered, when HbA1c is the primary variable of interest. No data are mentioned about patients' preferences or tolerance of drugs prescribed, only indirect information is available based on continuation of drugs delivered at pharmacies (filled prescriptions) or not. At the end health economy is mentioned. This is legitimate but has to be supported by an independent plan for analyses. Is the trial/study registered at clinicaltrials.gov or similar?
---

VERSION 1 – AUTHOR RESPONSE

No	Reviewer's comment	Author's response	Location in revised manuscript (page)
Reviewer 1			
	- The idea of the study is valid and sound because the propose to study the variables that may influence prescription of these 3 drugs besides metformin in UK.	Thank you for taking the time to review our manuscript.	
	- Exposure The exposure groups will include people prescribed SU, DPP4i, or SGLT2i as an add-on to metformin. Why not to include people with GLP1-RA? And insulin?	We chose not to include GLP1-RA and insulin as options for second-line therapy (first-stage intensification), as our pilot data and previous research showed low proportions of people prescribed these drugs as second-line antidiabetic treatment in the UK (~1-2% of all people initiating second-line treatment per year).¹ Although they mitigate unobserved confounding, IV estimators tend to be less efficient, requiring larger sample sizes as they use a subset of the variation in treatment assigned. For this reason, it is not feasible to include other second-line treatment options that are less commonly used in the UK. We have added this as a potential limitation in the summary section following the study abstract (page 5). ¹Wilkinson S, Douglas I, Stirnadel-Farrant H, et al. Changing use of antidiabetic drugs in the UK: trends in prescribing 2000–2017. BMJ Open. 2018;8(7):e022768.	5

	- Covariates: How economic status will impact in effectiveness? How will low-income people's prescription be addressed, considering higher costs for ISGLT2 and DPP4I? Will it be a bias?	Because the UK has a public health care system (the National Health Service), costs of drugs are not directly borne by the patient receiving the drug treatment. Therefore, we do not expect patients in our study to be directly influenced by drug prices. However, we are including the index of multiple deprivation (IMD) as a covariate in our analyses (details on page 11). This index ranks people into quintiles based on the income level of their postcode. We will use this as a proxy for socioeconomic status, which will allow us to account for some of the differences between exposure groups in terms of socioeconomic status. This issue may be relevant at the Clinical Commissioning Group (CCG) level, since some CCGs may recommend their clinicians to prescribe cheaper drugs as cost-saving measures. We will explore the variation in prescribing at the CCG level in our analyses, and check the IV balance across key covariates like IMD (proxy for socioeconomic status). Our pilot data showed considerable variation at the CCG level in second-line antidiabetic prescribing¹, which allows us to use prescribing patterns at the CCG level as our instrument in the instrumental variable analysis. ¹Wilkinson S, Douglas I, Stirnadel-Farrant H, et al. Changing use of antidiabetic drugs in the UK: trends in prescribing 2000–2017. BMJ Open. 2018;8(7):e022768.	
	Outcomes: - HbA1c varies a lot during 12 months. How many measurements will be considered per patient. Will you determine it prospectively?	Thank you for raising this important point. The sample size calculation is for our primary outcome, which is change in HbA1c at 12 months follow-up. However, we agree that we should consider HbA1c at other time points, as HbA1c can change in the medium-term. We have added HbA1c at 6, 18, 24, 30 and 36 as secondary outcomes of interest in the manuscript (page 12). Because we expect not all patients to have HbA1c measures at exactly 12 months	12, 17-18

		follow-up, we will use a 6-month window (e.g. 9-15 months follow-up), taking the HbA1c measure closest to 12 months from the start of follow-up for our primary outcome. In the case that HbA1c is missing during this 6-month time window, we plan to use linear interpolation to interpolate HbA1c at the outcome follow-up time point (e.g. at 12-months follow-up). This follows precedent research which used the same method to interpolate HbA1c values at specific follow-up points.¹ We have added this description under the new subheading “Missing data” on pages 17-18. While we are using historically collected data, we will only consider HbA1c measures recorded after initiating second-line treatment when investigating change in HbA1c, meaning that change in HbA1c will only be calculated using prospectively recorded HbA1c test results in relation to the initiation of second-line antidiabetic treatment. The baseline HbA1c measure we will use as the comparator will be the most recent HbA1c test result recorded prior to or on the day of second-line antidiabetic treatment initiation, within 180 days of the baseline date. We have amended the protocol to clarify that we will consider baseline HbA1c, eGFR, BMI, and blood pressure missing if the most recent test result is earlier than 180 days from the date of second-line initiation (page 12). As the reviewer points out, HbA1c (and these other measurements) vary over time. Those measures recorded earlier than 180 days before baseline are unlikely to be accurate at our study baseline (second-line treatment initiation). ¹Basu, A. et al. Development and Validation of the Real-World Progression in Diabetes (RAPIDS) Model. Medical decision making : an international journal of the Society for	
--	--	---	--

		Medical Decision Making 39 , 137-151, doi:10.1177/0272989X18817521 (2019).	
	- Cardiovascular outcomes need to be better defined. How an acute myocardial infarction or stroke will be registered? Will image methods be used? It is not clear if you will search for it prospectively or look into medical registry.	We have edited the outcomes section to make it clearer that we will use diagnosis codes from the hospital records (ICD-10 codes), as well as diagnosis codes from primary care (Read and Snomed codes) to define cardiovascular outcomes (pages 12-13). Unfortunately, imaging and other relevant test results (e.g. ECG) are not included in the electronic health record data sources we will use in this study. The Myocardial Infarction National Audit Project (MINAP) includes more detailed data (e.g. ECG and laboratory test results) related to myocardial infarction hospitalisations in England and Wales.¹ However, a previous study shows when linked with CPRD and HES (the datasets we will use in this project), CPRD and HES capture the majority of MI events also captured in MINAP in the general population.² Because this linkage is not routine and due to financial constraints, we cannot link our dataset to MINAP. We accept we may miss some incident (for outcomes) and prevalent (for covariates) cardiovascular events, which we expect to be non-differential in relation to the exposure. Consequently, our effect estimates may be somewhat diluted. Because our manuscript is already over the 4000-word limit set by the journal, we have not added this in our limitations section at this time. However, we believe this is an important limitation, and we plan to include it in our results manuscript where we describe the relative effects of these three second-line treatment options on cardiovascular disease outcomes. If the reviewer feels strongly this limitation should be mentioned in this protocol manuscript, we are happy to add it to the	12-13

		strengths and weaknesses section if the journal editors will allow. ¹https://www.nicor.org.uk/national-cardiac-audit-programme/myocardial-ischaemia-minap-heart-attack-audit/ ²Herrett E, Shah AD, Boggon R, et al. Completeness and diagnostic validity of recording acute myocardial infarction events in primary care, hospital care, disease registry, and national mortality records: cohort study. BMJ : British Medical Journal. 2013;346:f2350.	
	- Statistical methods should be better detailed for the baseline clinical and lab characteristics	Thank you – we have made edits to the covariates section of the manuscript to more clearly define the types of data we will use to define baseline clinical and laboratory characteristics. We have also made clear in this section that we will publish the codelists for all covariates we define alongside the study results. Briefly, we will use clinical diagnosis codes in primary care (Read or SNOMED codes) and in secondary care for the secondary-care linked sub-cohort (ICD-10 codes) to define comorbidities. In addition, we will use laboratory test results recoded in primary care to define relevant clinical measures such as blood pressure, estimated glomerular filtration (using serum creatinine), cholesterol levels, body weight, and body-mass index (BMI). Finally, we will use patient characteristics recorded in primary care (and secondary care for the secondary-care linked sub-cohort) to define age, ethnicity, sex. We have added the following baseline clinical and lab characteristics, after further consultation with health care professionals: history of cancer (any), ischaemic heart disease, proteinuria, prescriptions for renin-angiotensin system inhibitors (RASi) or statins within 60 days of second-line treatment initiation, general practice size, and time between baseline clinical/laboratory measures (BMI, HbA1c,	11-12, 17

		eGFR, body weight, blood pressure) and the second-line treatment initiation date. We have also separated our list of covariates into those we will adjust for in the primary analysis, and those we will additionally adjust for in the secondary analysis. We created a new subheading “Missing data” (page 17) in the analysis section to explain our rationale for splitting this list. In short, we anticipate higher levels of missing data for some of these covariates, and do not believe that using multiple imputation is possible since they are likely not missing at random and the IV model is computationally intensive. Therefore, we will conduct our complete case analysis in the primary analysis excluding the covariates with high levels of missing data, and conduct a secondary analysis with complete cases when including these additional covariates. This strategy has been used in previous high-impact observational research.¹ ¹Williamson, E. J. et al. Factors associated with COVID-19-related death using OpenSAFELY. Nature 584, 430-436, doi:10.1038/s41586-020-2521-4 (2020).	
	- The authors will use instrumental variable design method, a sophisticated method to account for unexpected behavior between variables, and to estimate relative treatment effectiveness.	Thank you for your comment.	
	- Sensitivity analysis are well described.	Thank you for your comment.	
	- Missing data are well addressed	Thank you for your comment.	17-18

		We have taken the opportunity to elaborate on our missing data section (pages 17-18), recognising that it is a complex issue with significant impacts on our study design and analysis. In short, we will not use multiple imputation because the IV model is very computationally intensive, and we cannot assume all covariates/outcomes will be missing at random. Therefore, we will use complete case analysis using a main set of covariates, and additionally adjust for further covariates expected to have a lot of missing data in a secondary analysis, again with complete case approach. For outcomes, we will use linear interpolation to fill in missing outcome at particular follow-up time points, or inverse probability weighting where outcome data are missing due to loss to follow up. Further details are provided in the missing data section on pages 17-18 of the manuscript.	
	- A concern is how to deal with patients who may change from one group to another (eg when a certain medication is momentarily out in the market or may not be available and the physician just change).	We will use an intention-to-treat approach when analysing our data. This follows methods used in observational research which seeks to emulate randomised controlled trials.¹ We accept that some people may switch drug treatments during follow-up permanently or even temporarily, most likely during periods of acute illness or if treatment is intensified/switched to better manage blood glucose. Our future work, discussed briefly on page 20, will consider treatment switching and time-updated confounding during follow-up. We have made clear that we will publish the methods and analysis plan for this work at a later date (page 20). ¹Hernán, M. A. & Robins, J. M. Using Big Data to Emulate a Target Trial When a Randomized Trial Is Not Available. Am J	20

		Epidemiol 183 , 758-764, doi:10.1093/aje/kwv254 (2016).	
	- A major concern is that T2DM patients generally receive SU when in a more advanced T2DM stage with longer duration, when they become more insulinopenic. With a longer duration. In these patients, however, we also expect a higher incidence of cardiovascular disease. By this way, observational studies may face a reverse causality effect, falsely increasing cardiovascular risk for SU. How to deal with this?	We agree with the reviewer – people prescribed Metformin-SU combination therapy likely have higher HbA1c and certain risk factors which encourage the clinician to prescribe SU as the second-line treatment choice. However, by using the variation in prescribing at the Clinical Commissioning Group (CCG)-level, we hope to minimise this confounding by indication. We agree that for many observational studies, reverse causality is a concern. The IV method addresses this since for a given patient we do not anticipate that whether their CCG has a higher tendency to prescribe a particular treatment would be strongly related to that patient’s diabetes duration/insulinopenic status. Therefore using this variation is sufficient to break the ‘reverse causality’ link.	
Reviewer 2			
	The paper is of great clinical interest presenting valuable data. However, the below proposed revisions may significantly increase visibility and overall quality of this important publication.	Thank you for taking the time to review our manuscript.	
	1. Keywords should be extended presenting items highly relevant for the study which would attract more attention of multi-professional groups. Following items might be helpful: Patient stratification,	Thank you for your suggestion. We have added the following key terms on our title page in order to reach a wider audience: treatment choice, comparative effectiveness, pharmacoepidemiology. We hope the reviewer agrees these terms are appropriate.	4

	disease modelling, targeted treatment prediction.	Unfortunately, we are limited in the key terms we are able to attach to the submission in the BMJ Open submission portal by a drop-down menu of pre-defined key terms. We are happy to add the key terms we list on our title page to the submission if the journal editors can allow.	
	2. Statements in the "Introduction" should be updated keeping in mind a broad spectrum of T2DM related secondary pathologies and corresponding economic burden to healthcare. In particular, the 3rd sentence should be completed such as (just a proposal) "...careful management of blood glucose and cascading pathologies". Corresponding references should be provided - see some suggestions presented under the below point 4.	Thank you for this suggestion. We have edited this sentence, which now reads "...careful management of blood glucose and diabetes-associated complications." We hope this addresses the reviewer's concern. We have cited the NICE (National Institute for Health and Care Excellence) guidance for diabetes management which is most relevant to our UK setting. This guidance describes the diabetes-associated complications from which people are at risk with poor diabetic control. We have also added a reference to a meta-analysis published in 2021 in the introduction section, as we agree with the concerns brought up by the reviewer in comment 2 and 4 that references should be updated. More details are provided in the response to the reviewer's comment 4.	6
	3. Legends should be essentially extended to detail the clinical relevance of the presented data.	Thank you for your suggestion – we agree with the reviewer that the captions were not descriptive enough to explain the relevance of the figures presented. We have updated the legends for Figures 1-3 to better describe the clinical relevance of the information we are presenting.	Figures 1-3 captions
	4. References should be essentially updated: in the current version of the manuscript only 2 references originated from the year 2020 and no one from the year 2021. Below proposed items and corresponding references might be useful to fulfil the task	Thank you for these recommendations. We agree this manuscript would benefit from updated references. As we only submitted this manuscript to BMJ Open November 2020, we could only include references which had been published by that date. We have added a citation to a meta-analysis published in 2021 which evaluated the effectiveness of the drug treatments we intend to study.¹ This reference is listed as	Page 7, reference 15

	by completing "Introduction", "Discussion", "Future work" as well as "Abstract": A. Individualised Prediction and Targeted Prevention in diabetes care:  - Nomogram prediction for the 3-year risk of type 2 diabetes in healthy mainland China residents. 2019. doi: 10.1007/s13167-019-00181-2. - Suboptimal health status as an independent risk factor for type 2 diabetes mellitus in a community-based cohort: the China suboptimal health cohort study. 2019. doi: 10.1007/s13167-019-0159-9. B. Tools for companion diagnostics, disease prediction and prognosis (Big data, multi-omics, AI, Machine learning, Biobanks):  - Cell-free nucleic acid patterns in disease prediction and monitoring-hype or hope? 2020. doi: 10.1007/s13167-020-00226-x. - Biobanks in the era of big data: objectives, challenges, perspectives, and innovations for predictive, preventive, and personalised medicine. 2020. doi: 10.1007/s13167-020-00213-2 	number 15 in our references section. This study found differences in HbA1c control; however, was unable to stratify these results by patient characteristics such as age, sex, ethnicity, etc. We hope our study will be able to provide useful new data to allow health care teams and patients select the best treatment based on individual characteristics. ¹Mannucci E, Naletto L, Vaccaro G, Silverii A, Dicembrini I, Pintaudi B, et al. Efficacy and safety of glucose-lowering agents in patients with type 2 diabetes: A network meta-analysis of randomized, active comparator-controlled trials. Nutrition, Metabolism and Cardiovascular Diseases. 2021;31(4):1027-34.	
--	--	---	--

	- Artificial intelligence supported patient self-care in chronic heart failure: a paradigm shift from reactive to predictive, preventive and personalised care. 2019. doi: 10.1007/s13167-019-00188-9. - Evaluation of machine learning methodology for the prediction of healthcare resource utilization and healthcare costs in patients with critical limb ischemia-is preventive and personalized approach on the horizon? 2020. doi: 10.1007/s13167-019-00196-9.		
	5. Keeping in mind the above listed points, "Abstract " should be thoroughly elaborated presenting A. DM relevant UK / global statistics which would clearly support the study objectives, design and conclusions B. concluding statements in the context of predictive and personalised medicine to improve DM related primary, secondary and tertiary care C. outlook (predictive models, AI, cost-efficacy, improved life quality of the patients).	Thank you for your suggestions. Unfortunately, our abstract is already at the 300-word limit set by the journal. Considering the limited space available in the abstract, we would prefer to keep the focus on describing the three second-line antidiabetic treatment options we will compare, the instrumental variable methods, and study outcomes. We agree with the reviewer that it is important to describe the relevance of diabetes and personalised medicine in our study setting. Our introduction begins with diabetes statistics relevant to the UK, including prevalence and cost-share to the England National Health Service. We have also added a sentence which highlights the persistent uncertainty in how to best tailor second-line antidiabetic treatments to people with diabetes (page 7). The sentence reads: "However, there is still uncertainty among regulators, clinicians, and patients on how best to tailor second-line antidiabetic treatment based on individual characteristics."	7

Reviewer 3			
	This study aims to evaluate the relative effectiveness of the three most common second-line medications added on to the first-line metformin in the UK. Currently SU, DPP4i and SGLT2i are added based on the physician discretion/preference in individuals with type 2 diabetes who do not reach glycaemic target on metformin alone and this study aims to improve the tailoring of the added medication to the participant's risk profile using the Instrumental Variable method.	Thank you for taking the time to comment on our article.	
	Overall, I feel that I did not fully understand how will this study improve the treatment of individuals with type 2 diabetes. Do the authors suggest that the IV method will be superior to the current method (physician preference)? Apologies if I got it all wrong, but as someone who reads the literature and researches this area themselves, it probably means that further details and clarifications should be added.	We have made an addition to the introduction (page 7) to highlight the gap that persists in the evidence base for how exactly to tailor second-line antidiabetic medication based on individual characteristics. The new sentence reads: "However, there is still uncertainty among regulators, clinicians, and patients on how best to tailor second-line antidiabetic treatment based on individual characteristics." We recognise that clinicians have significant expertise in treating type 2 diabetes, and their treatment choices are informed by evidence from clinical trials, observational research, and their own clinical experience. However, randomised controlled trials have not yet directly compared the effectiveness of combination therapy with metformin and SU, DPP4i, or SGLT2i, particularly among subgroups of patients sharing particular characteristics (e.g. ethnicity, age).	7

		There is therefore great interest in using observational data to compare these three drug treatments overall and by clinically important subgroups, in the absence of randomised controlled trial data, in order to generate evidence which can be applied by clinicians' and patients' when making treatment decisions. Confounding by indication is a major limitation of observational pharmacoepidemiology research. We aim to use the instrumental variable method to reduce confounding by indication when comparing the three second-line treatments of interest to produce meaningful results from our observational dataset, overall and across clinically important subgroups. We hope these results will then help drug regulators (e.g. NICE in England), patients, and clinicians choose the optimal second-line antidiabetic treatment based on individual characteristics.	
	Methods 1. Please clarify how the clinically important subgroups will be determined. Are they predefined based on the baseline data?	On page 17, we pre-specify the following subgroups, which were identified based on literature review, and clinical and patient knowledge imparted by the grant holders:  - People with and without cardiovascular comorbidities - People with baseline eGFR above and below 60 mL/min/1.73m² - Age groups - Sex - Ethnicity - BMI categories - Baseline HbA1c levels - Adherence to metformin We have made an edit on page 17 to clearly state that any additional subgroups of interest, beyond those that we state in this paper, will be informed by descriptive statistics and the advice of health care professional panels based on the baseline data. The sentence reads: "Any additional subgroups will be informed by descriptive	17

		statistics of each covariate, and the advice of a panel of health care professionals, building on those identified in a literature review. The clinical panelists will include diabetologists, GPs, and practice nurses involved in care for people with T2DM.” We hope that our sample size will allow us to investigate additional subgroups and possibly interactions between more than one subgroup. We will evaluate if there are sufficient numbers of people included in our cohort to add additional subgroup analyses using the baseline characteristics. We hope the edit to the manuscript has made it clear to the reviewer how we will select our subgroups of interest.	
	2. The choice to include individuals as young as 18 year old, would limit the possibility to assess the micro- and macro-vascular benefits of the treatment. Please explain why you did not pick an older cut-off.	We chose not to select an older cut-off since this would limit the generalisability of our study. Furthermore, type 2 diabetes incidence is increasing in lower age groups, particularly in ethnic minority groups in the UK.¹ We follow precedent research, both randomised-controlled trials and observational studies, which has included people 18 years or older.²⁻⁵ We hope that our large sample size may allow us to observe differences in treatment effect in younger age groups, even where incidence of certain outcomes is expected to be quite low. An important pre-specified stratifying variable will be age group, in which we may see greater relative treatment effects in the older age groups. ¹https://www.diabetes.org.uk/resources-s3/2017-11/diabetes_in_the_uk_2010.pdf	

		²Wilkinson, S. et al. Changing use of antidiabetic drugs in the UK: trends in prescribing 2000–2017. BMJ Open 8, e022768, doi:10.1136/bmjopen-2018-022768 (2018). ³Wilkinson, S. et al. Factors associated with choice of intensification treatment for type 2 diabetes after metformin monotherapy: a cohort study in UK primary care. Clin Epidemiol 10, 1639-1648, doi:10.2147/CLEP.S176142 (2018). ⁴Leiter, L. A. et al. Canagliflozin provides durable glycemic improvements and body weight reduction over 104 weeks versus glimepiride in patients with type 2 diabetes on metformin: a randomized, double-blind, phase 3 study. Diabetes care 38, 355-364, doi:10.2337/dc13-2762 (2015). ⁵Cefalu, W. T. et al. Efficacy and safety of canagliflozin versus glimepiride in patients with type 2 diabetes inadequately controlled with metformin (CANTATA-SU): 52 week results from a randomised, double-blind, phase 3 non-inferiority trial. Lancet (London, England) 382, 941-950, doi:10.1016/s0140-6736(13)60683-2 (2013).	
	3. The inclusion of individuals treated with metformin from 2011 to 2020 means that some will have longstanding diabetes and some will be recently diagnosed. Is this a limitation?	We have edited the study population section and figure 1 (flow diagram) as we were unclear in the original manuscript exactly how our cohort was selected. We require individuals in our cohort to have both a metformin monotherapy prescription and another type of antidiabetic prescription between 2011-2020 (rather than just a metformin monotherapy prescription, as was originally written). We then extract the entire prescribing history of everyone with both metformin monotherapy prescription and any other antidiabetic prescription between 2011-2020. We use the entire prescribing history to then determine when an individual	Page 9, Figure 1

		initiated first-line treatment, and when they initiate second-line. We decided to restrict the study time period to 2014 due to SGLT2i not being widely prescribed before then part way through the data extraction process- hence why we identify people who intensify between 2011-2013 who are later excluded. We will include the time on metformin monotherapy as a covariate in our analysis and ensure this is balanced across our exposure groups, and makes sense clinically with the investigators who are diabetologists. We hope this information makes it clear that we should have a representative sample of the UK diabetic population who initiate second-line treatment.	
	4. Regarding the sample size/power calculation: a. Why does Table 1 not include the sample size for the DPP4i? b. Please explain why the sample size (27,000) is several folds higher than the minimum required (for 80% power and effect size of 0.4% reduction, Table 1). Is this to account for expected loss to follow up?	a. We chose to focus on the two treatment groups expected to have the lowest number of participants (SU and SGLT2i), since these groups will limit the statistical power available to determine a true difference in treatment effect. Statistical power when comparing either SU or SGLT2i to DPP4i (the exposure group expected to have the largest number of people) will be greater than the SU v SGLT2i comparison, since these are expected to be the smallest exposure groups. b. We will use the maximum sample available to us in the Clinical Practice Research Datalink (CPRD) dataset. This is to maximise statistical power for our primary analysis, and allow us to stratify by clinically important subgroups. Our feasibility count suggests we will have more people in our study than the minimum required to observe clinically important differences in HbA1c at 1 year follow-up (primary outcome) with	

		various IV compliance levels. This is important as we are likely to be missing some HbA1c measures at 12 months follow-up for some participants or have some people censored before 12 months follow-up, as the reviewer rightly points out.	
Reviewer 4			
	This is description of the design of an observational study comparing the prescription patterns of second-line drugs as add-on to metformin in patients with type 2 diabetes in UK primary health care. Even if the design is purely observational, new statistical methods (instrumental variable = local recommendations by authorities) are introduced to minimize the risk of confounding by indication. Three classes of second-line anti-diabetes drugs will be analysed for prescription patterns during 2014-2020 in different regions of the UK (SU, DPP-4 inhibitors; SGLT-2 inhibitors), but not GLP-1 receptor agonists/analogues because much less prescription in clinical practice during early years.	Thank you for taking the time to review our manuscript.	
	I think that the aim and methods of this observational study are mostly satisfactory, but a first focus on control of HbA1c over a 12	Thank you for your comments. We appreciate that clinical outcomes in particular cardiovascular and renal, are more important than surrogate markers such as HbA1c. We agree that the earliest clinical signs of microvascular damage to the	12-13

	months period is not enough. What matters is the potential to prevent hard clinical endpoints (CVD and renal) as well as mortality patterns and hospitalisations (available in HES for 70% of all hospitalisations) during a 6 year period, as HbA1c is more linked to microvascular endpoints than to macrovascular endpoints. It is not clear whether prevention of nephropathy is included as an endpoint or not, just "microvascular endpoints" are mentioned. The authors state that they will consider ESRD, and 40% decline in eGFR from baseline. However, nephropathy is not explicitly mentioned.	kidneys, new onset nephropathy, is an important outcome. In protocol development we did not feel that recording of proteinuria in routine data was adequate for this to be a reliable outcome and therefore focused on substantial changes in eGFR or commencement of renal replacement therapy, both of which we knew to be well recorded in routine care data. However, early developmental work has suggested that about 58% of people have recorded albumin: creatinine ratio within 2 years of drug start. Therefore, we have added an additional renal outcome to the protocol (Page 12-13): “Outcomes for long-term relative effectiveness (objective 3) will include macro- and micro-vascular conditions such as cardiovascular outcomes (MI, CHF, unstable angina, stroke), renal outcomes (nephropathy, ESRD, 40% decline in eGFR from baseline(29)), and lower limb amputation. ... We will define nephropathy as new onset albuminuria or eGFR<60mL/min/1.73m2 in people with eGFR≥60mL/min/1.73m2 and no raised albumin to creatinine ratio within two years of second-line treatment initiation.” We hope the reviewer agrees this is an appropriate outcome to add to our analysis.	
	A number of covariates are listed, but not including BMI. There might exist a bias in prescription patterns according to BMI of the patient. It is well-known that SU will increase hyperinsulinaemia as often found in obese patients with underlying insulin resistance. Why is not BMI considered? Only as a secondary outcome at 12 months is BMI considered,	We have edited the covariate section of the manuscript as it was not clear enough how we would use the data to identify our covariates of interest (page 11-12). Indeed, BMI is an important covariate and will be included in our analyses. We hope the edits to this section have made it clear we will include BMI as a covariate in our analyses.	11-12

	when HbA1c is the primary variable of interest.		
	No data are mentioned about patients' preferences or tolerance of drugs prescribed, only indirect information is available based on continuation of drugs delivered at pharmacies (filled prescriptions) or not.	We agree that this is a limitation of our dataset. We are limited in our prescribing data, and can only assume that a patient continues their treatment and takes their treatment as prescribed based on prescriptions ordered by their GP. Because our manuscript is already over the 4000-word limit set by the journal, we have not added this in our limitations section at this time. However, we intend to discuss this limitation in future papers describing the results from this study. If the reviewer feels strongly this limitation should be mentioned in this protocol manuscript, we are happy to add it to the strengths and weaknesses section if the journal editors will allow.	
	At the end health economy is mentioned. This is legitimate but has to be supported by an independent plan for analyses	We agree. We have made an edit on page 20 to indicate we will publish the analysis plan for the health economic analyses separately (page 20). We could not fit the methods for this analysis in the current manuscript given the word limit of the journal.	20
	Is the trial/study registered at clinicaltrials.gov or similar?	We have not registered our study at clinicaltrials.gov as it is an observational study using routinely collected historical health data. By publishing this protocol, we hope to clearly set out our analysis plan and refer back to this publication in subsequent manuscripts. We will also make use of our study website, where we will post links to published code/codelists, study updates, analysis plans, etc.	

		https://www.lshtm.ac.uk/research/centres-projects-groups/permit	
--	--	---	--

VERSION 2 – REVIEW

REVIEWER	Golubnitschaja, Olga Friedrich-Wilhels-University of Bonn
REVIEW RETURNED	01-Aug-2021

GENERAL COMMENTS	The revision has somewhat improved the paper quality. However, the authors could make better use of recommendaions provided by reviewers,
---

REVIEWER	Samocha-Bonet, Dorit Garvan Institute of Medical Research, Diabetes Division
REVIEW RETURNED	03-Aug-2021

GENERAL COMMENTS	The Authors have addressed all my comments, thank you.
--

REVIEWER	Nilsson, Peter Lund University, Clinical Sciences
REVIEW RETURNED	04-Aug-2021

GENERAL COMMENTS	The manuscript has improved. I guess that this study plan is as far as one can get based on observational analyses in the field of slecting second-line oral drugs for type 2 diabetes, but evidence from RCT´s must always be acknowledged and respected. This is escpecially relevant for the SGLT2-inhibitors that have been proven to be very successful vs. placebo for CVD and nephropathy prevention in clinical trials dominated by secondary prevention and fulfilling FDA requirements. The DPP-4 inhibitors lack this kind of evidence as do also the SU class of drugs generally speaking. It is not enough to state that SU lack risk of CVD (based on meta-analyses), the point is to show clinical benefits for CVD prevention. This is lacking for SU.
--

VERSION 2 – AUTHOR RESPONSE

No	Reviewer's comment	Author's response	Location in revised manuscript (page)
Reviewer 2	The revision has somewhat improved the paper quality. However, the authors could make better use of recommendaions provided by reviewers,	Thank you for taking the time to review our manuscript. We appreciated all comments submitted by the peer reviewers, which we believe have improved our manuscript substantially. We did our best to address all comments, while also keeping to	

		journal requirements (e.g. word count limits, specific key terms available).	
Reviewer 3	The Authors have addressed all my comments, thank you.	Again, thank you for taking the time to review our manuscript. The comments were important to helping us improve the quality of this work.	
Reviewer 4	The manuscript has improved. I guess that this study plan is as far as one can get based on observational analyses in the field of selecting second-line oral drugs for type 2 diabetes, but evidence from RCT's must always be acknowledged and respected. This is especially relevant for the SGLT2-inhibitors that have been proven to be very successful vs. placebo for CVD and nephropathy prevention in clinical trials dominated by secondary prevention and fulfilling FDA requirements. The DPP-4 inhibitors lack this kind of evidence as do also the SU class of drugs generally speaking. It is not enough to state that SU lack risk of CVD (based on meta-analyses), the point is to show clinical benefits for CVD prevention. This is lacking for SU.	Again, thank you for taking the time to review our manuscript. We agree that appraisal and consideration of RCT results is critical to interpreting the results of our observational study. We plan to conduct a literature review of published RCTs which compare the relative effectiveness of any of our antidiabetic treatments of interest. We have edited the manuscript to make this explicit, adding it to "Objective 1" of the protocol. The section now reads (with new additions bolded): Objective 1: Describing UK treatment patterns for second-line T2DM treatment, and summarising the results of relevant published RCTs to contextualise the study findings. We will describe trends in prescribing for T2DM second-line treatment for the duration of the study period across the UK and between CCGs. This analysis will update previous research which described the same second-line treatment use in the UK from 2000 to 2017, and will employ similar methods.(4) These descriptive statistics will inform the assessment of the validity of the assumptions that underlie the	13-14

		overall study design. Baseline characteristics listed in the covariates section will also be described for this cohort, overall and stratified by exposure group. We will also conduct a literature review to summarise published RCTs which describe the relative effectiveness of alternative second-line antidiabetic treatments of interest to this study. This will help contextualise the results of this observational study (cf objectives 2 and 3). We will consider reasons for any possible differences between this observational study compared to published RCTs, including residual confounding and differences in the study populations. ... We hope the reviewer finds this edit satisfactory.	
--	--	--	--

VERSION 3 – REVIEW

REVIEWER	Nilsson, Peter Lund University, Clinical Sciences
REVIEW RETURNED	29-Aug-2021
GENERAL COMMENTS	The manuscript has improved and I am glad that the authors state that the any outcomes of their observational study will be discussed within the context of existing RCT's using antidiabetes drugs for CVD prevention.